# Is the Ventilatory Efficiency in Endurance Athletes Different?—Findings from the NOODLE Study

**DOI:** 10.3390/jcm13020490

**Published:** 2024-01-16

**Authors:** Przemysław Kasiak, Tomasz Kowalski, Kinga Rębiś, Andrzej Klusiewicz, Maria Ładyga, Dorota Sadowska, Adrian Wilk, Szczepan Wiecha, Marcin Barylski, Adam Rafał Poliwczak, Piotr Wierzbiński, Artur Mamcarz, Daniel Śliż

**Affiliations:** 13rd Department of Internal Medicine and Cardiology, Medical University of Warsaw, 02-091 Warsaw, Poland; 2Department of Physiology, Institute of Sport—National Research Institute, 01-982 Warsaw, Poland; 3Department of Physical Education and Health in Biala Podlaska, Branch in Biala Podlaska, Jozef Pilsudski University of Physical Education, 00-968 Warsaw, Poland; 4Department of Kinesiology, Institute of Sport—National Research Institute, 01-982 Warsaw, Poland; 5Department of Internal Medicine and Cardiac Rehabilitation, Medical University of Lodz, 90-419 Lodz, Poland

**Keywords:** prediction equation, cardiac physiology, cardiopulmonary exercise testing, VE/VCO_2_-slope, cardiorespiratory fitness, exercise ventilation

## Abstract

**Background:** Ventilatory efficiency (VE/VCO_2_) is a strong predictor of cardiovascular diseases and defines individuals’ responses to exercise. Its characteristics among endurance athletes (EA) remain understudied. In a cohort of EA, we aimed to (1) investigate the relationship between different methods of calculation of VE/VCO_2_ and (2) externally validate prediction equations for VE/VCO_2_. **Methods:** In total, 140 EA (55% males; age = 22.7 ± 4.6 yrs; BMI = 22.6 ± 1.7 kg·m^−2^; peak oxygen uptake = 3.86 ± 0.82 L·min^−1^) underwent an effort-limited cycling cardiopulmonary exercise test. VE/VCO_2_ was first calculated to ventilatory threshold (VE/VCO_2_-slope), as the lowest 30-s average (VE/VCO_2_-Nadir) and from whole exercises (VE/VCO_2_-Total). Twelve prediction equations for VE/VCO_2_-slope were externally validated. **Results:** VE/VCO_2_-slope was higher in females than males (27.7 ± 2.6 vs. 26.1 ± 2.0, *p* < 0.001). Measuring methods for VE/VCO_2_ differed significantly in males and females. VE/VCO_2_ increased in EA with age independently from its type or sex (β = 0.066–0.127). Eleven equations underestimated VE/VCO_2_-slope (from −0.5 to −3.6). One equation overestimated VE/VCO_2_-slope (+0.2). Predicted and observed measurements differed significantly in nine models. Models explained a low amount of variance in the VE/VCO_2_-slope (R^2^ = 0.003–0.031). **Conclusions:** VE/VCO_2_-slope, VE/VCO_2_-Nadir, and VE/VCO_2_-Total were significantly different in EA. Prediction equations for the VE/VCO_2_-slope were inaccurate in EA. Physicians should be acknowledged to properly assess cardiorespiratory fitness in EA.

## 1. Introduction

Ventilatory efficiency (VE/VCO_2_) describes the relationship between pulmonary ventilation (VE) and carbon dioxide production (VCO_2_) [1]. To keep acid–base balance during exercise, VE grows simultaneously with VCO_2_ [2]. VE/VCO_2_ merges circulatory and respiratory functions [1,3]. Several ways of measuring VE/VCO_2_ have been proposed including VE/VCO_2_ from start to first ventilatory threshold (VT1) (VE/VCO_2_-slope), as the minimal continuous value (VE/VCO_2_-Nadir) and across whole physical effort (VE/VCO_2_-Total) [4]. 

Professional and elite athletes are exposed to high training loads [5]. Thus, more endurance athletes can be referred to with suspected cardiovascular diseases (CVD) [5]. Knowledge of cardiorespiratory indicators among endurance athletes remains important [6]. Precise risk stratification is of critical importance to ensure safe physical activity because it enables medical professionals to adjust clinical management and exercise intensity [5]. An important prognostic role in CVD has been assigned to the VE/VCO_2_-slope [7]. VE/VCO_2_-slope is elevated in pulmonary hypertension, chronic obstructive pulmonary disease, interstitial lung disease, and other diseases. The pivotal role of the VE/VCO_2_-slope was noted in heart failure (HF) [2,8]. 

Despite the VE/VCO_2_-slope being a submaximal parameter, its prognostic power is comparable to peak oxygen uptake (VO_2_peak) [1,9]. Studies on untrained populations reported lower VE/VCO_2_ measurements in males and its increase with aging [10]. During clinical assessment, a cutoff below 30 is considered normal, and values above 34 to 36 suggest a high risk of mortality [7,11,12]. 

Previous research focused mainly on univariable measurements [13,14]. VE/VCO_2_ is a ratio of two variables, i.e., VE and VCO_2_. In other words, it is calculated by dividing one parameter by another. VE grows simultaneously with VCO_2_ during continued physical effort to ensure efficient excretion of the produced metabolites [15]. The link between VE and VCO_2_ is multifactorial. The slope continuously and stably increases from the start to VT1 [16]. Above this, VE is forced by lactate accumulation and the slope begins to steepen [10]. However, the physiology of VE/VCO_2_ in endurance athletes is understudied.

Prediction equations provide several benefits. They allow for indirect calculation and facilitate the determination of the participant’s health based on the comparison of direct measurements with predicted reference values [13]. There were some attempts to predict VE/VCO_2_-slope with regression models [8,10,17,18,19,20], but there is no validation of those models. Previous research suggests that the usage of unified equations might not be optimal both in untrained individuals and athletes [8,21]. Few studies evaluated VE/VCO_2_ in athletes [22,23,24]. However, none of them verified the underlying dependency of different VE/VCO_2_ measurements. Potentially, well-trained endurance athletes could maintain strenuous physical effort well beyond VT1, where VE/VCO_2_-slope increases nonlinearly [25]. The use of inaccurate prediction models has negative consequences. Underestimated or overestimated values may lead to incorrect monitoring of training and disregard of potential risk factors. In turn, overestimated values may unnecessarily increase awareness and prevent demanding physical activity [3,13,21].

Research suggests that in athletes, VE/VCO_2_ could be independent of the type of exercise testing, body mass, or endurance capacity [23]. Based on our recent studies on well-trained individuals [14,21], we stipulate that current prediction models do not allow for a transferable calculation of cardiorespiratory parameters from the general population. Moreover, the underlying relationship between several measuring options for VE/VCO_2_ in athletic cohorts is still controversial. This study aimed to (1) externally validate prediction equations for the VE/VCO_2_-slope in the athletic population and (2) explain the relationship between the VE/VCO_2_-slope, VE/VCO_2_-Nadir, and VE/VCO_2_-Total.

## 2. Materials and Methods

### 2.1. Study Design

This study was conducted following the guidelines of the EQUATOR Network for observational studies: Strengthening the Reporting of Observational Studies in Epidemiology (STROBE) Statement [26]. The STROBE Checklist for Cross-Sectional Studies is provided in the Appendix A. Research has been reviewed and approved by the Bioethics Committee of the Medical University of Warsaw (AKBE/277/2023). Participants provided their written informed consent. All study procedures were in line with the Declaration of Helsinki. 

Healthy endurance athletes were referred to the standardized CPET. An endurance athlete was defined as having at least four years of regular training and competitive experience (at local and international levels). Participants held a membership in a sports association or training club and were also members of elite or development national teams. The tests were carried out in the years 2022–2023 at the Institute of Sport—National Research Institute in Warsaw (https://insp.pl; accessed on 22 October 2023). To be included, all participants underwent a medical evaluation by a physician (physical examination, medical and family history) according to the routine procedures of the testing center.

We applied a rigorous selection process to obtain a group free of disturbing factors to test the raw relationships of VE/VCO_2_ and ensure safe exercise tests. The preliminary exclusion criteria were any of the following: (1) respiratory diseases, (2) CVD, (3) neurological and psychiatric conditions, (4) musculoskeletal injuries limiting performance during CPET, (5) deviations in complete blood count, and (6) being a smoker. The final selection criteria were (1) age ≥ 18 years and (2) maximum CPET. For a visual presentation of the recruitment procedures, see Figure 1, and for exact definitions of exclusion criteria, see the Appendix A.

### 2.2. CPET Protocol

CPETs were conducted following the same unified protocol. All tests were performed under unified laboratory conditions. According to the reference values for CPET in endurance athletes, the maximal effort was confirmed by the following: (1) respiratory exchange ratio (RER) ≥ 1.05, (2) plateau in VO_2_ (stable VO_2_ for ≥30 s), (3) volitional inability to maintain effort, (4) Borg rating of perceived exertion ≥ 18, and (5) maximal heart rate (HR) ≥ 80% of age-predicted [13,27]. The physiologist supervised each CPET. Participants were verbally encouraged to achieve peak performance.

CPET was performed on an upright cycle ergometer (Cyclus II Ergometer, RBM, Leipzig, Germany) in a ramp protocol. The tests began with a 2–3 min warm-up in the form of light pedaling without resistance. Participants completed an incremental test starting from 55 to 70 W, gradually increasing the load by 0.17–0.28 W·s^−1^. The working loads were individually adjusted in all provided ranges according to the performance capabilities of each athlete. 

### 2.3. Study Endpoints

We measured basic demographic data: sex, age, height, weight, body mass index (BMI), and exercise performance. We obtained weight with the usage of a TANITA device (TANITA Corporation, Arlington Heights, IL, USA) and height with the usage of a SECA stadiometer (SECA GmbH & Co., Hamburg, Germany). Both weight and height were measured in the morning before breakfast. HR was measured with the Polar H10 chest strap (Polar Electro Oy, Kempele, Finland), continuously synchronized with the Cortex B3 Metamax. VE, VCO_2_, oxygen uptake, respiratory rate, and tidal volume were measured by the Cortex B3 Metamax using the breath-by-breath method (Hans Rudolph V2 Mask, Hans Rudolph, Inc., Shawnee, KS, USA). Variables were averaged in 15-s intervals. All measurement devices were calibrated for each usage in line with the producer’s instructions. 

The v-slope method has been previously used to find VT1 [28]. VT1 was identified in all endurance athletes enrolled in this study. VE/VCO_2_-slope was defined as the linear relationship between VE and VCO_2_ from the start to VT1, excluding the first minute of the protocol, where noise values emerged. VE/VCO_2_-Nadir was defined as the lowest continuous 30 s average. VE/VCO_2_-Total was calculated across the whole CPET protocol without the first minute of the protocol.

### 2.4. Sample Characteristics

In total, 140 healthy, well-trained individuals fulfilled the study criteria. Sample characteristics stratified by sex are presented in Table 1. There were 77 (55.0%) males and 63 (45.0%) females. Participants represented the following endurance sports: 56 (40.0%) trained triathlon or cycling, 59 (42.1%) chose speedskating, and 25 (17.9%) preferred other disciplines. VO_2_peak was 3.21 ± 0.48 L·min^−1^ for females and 4.40 ± 0.64 L·min^−1^ for males. Females noted a higher VE/VCO_2_-slope than males (27.7 ± 2.6 vs. 26.1 ± 2.0). Two (2.6%) males exceeded the cutoff of 30 both for VE/VCO_2_-slope and VE/VCO_2_-Total, while all males maintained normal VE/VCO_2_-Nadir. Eleven (17.5%) females exceeded the cut off of 30 both for VE/VCO_2_-slope and VE/VCO_2_-Total. Four (6.3%) of them had VE/VCO_2_-Nadir above 30. There were no males with any VE/VCO_2_ > 34; however, it was present for 4 (6.3%) females in VE/VCO_2_-Total. 

### 2.5. Selection of Prediction Models for Validation

Prediction equations for the VE/VCO_2_-slope were collected based on Paap and Takken reference values for CPET [29,30] and by additional searches in 5 databases: PubMed, Web of Science, Scopus, Google Scholar, and Embase. Applied keywords were “ventilatory efficiency”, “VE/VCO_2_-slope”, “prediction model”, “prediction equation”, “reference values”, and “linear regression”. To ensure similarity with our group, we excluded models primarily derived from pediatric or geriatric populations (<18 or >70 years old) and clinical samples with coexisting medical conditions. Finally, 12 models from 6 studies met the selection criteria, and their detailed description is presented in Table 2. All selected equations predicted the VE/VCO_2_-slope to the first ventilatory threshold. 

### 2.6. Statistical Analysis

Data distribution was examined by the Shapiro–Wilk test and quantile–quantile plots. Due to parametric distribution, continuous variables are presented as mean ± standard deviation. Categorical variables are presented as numbers (percentages). In cases of missing data in any of the measured variables, participants were excluded from the analysis to ensure maximum precision. The sample was evaluated in the G*Power Software (version 3.1.9.6) [31] to obtain significance (*p* < 0.05) and large effect sizes for each applied statistical test. All achieved statistical powers were >0.8. 

Differences between VE/VCO_2_-slope, VE/VCO_2_-Nadir, and VE/VCO_2_-Total or observed and predicted VE/VCO_2_-slope were calculated using the Student *t*-test or Wilcoxon test, as appropriate. The precision of the equations was examined by the mean absolute percentage error (MAPE) and root mean square error (RMSE). RMSE was adjusted to percentage by dividing the errors by the mean of observed VE/VCO_2_-slope (%RMSE). Two-way mixed effects interclass correlation coefficients (ICC_3,1_) with 95% confidence intervals (CI) [32,33] were calculated to test agreement between the observed and predicted VE/VCO_2_-slope. The relationship between observed and predicted values was visually presented by Bland–Altman plots. We regressed the predicted VE/VCO_2_-slope against direct measurements and presented it by the coefficient of determination (R^2^). 

Data are presented following the AMA *Manual of Style*. A two-sided *p*-value < 0.05 was considered as significant. Analyses were performed in the IBM SPSS Statistical Software (version 29.0, IBM, Chicago, IL, USA). Figures were derived via GraphPad Prism (version 10.1, GraphPad Software, San Diego, CA, USA).

## 3. Results

### 3.1. Interdependency of VE/VCO_2_ Measurements

Relationships between the VE/VCO_2_-slope, VE/VCO_2_-Nadir, and VE/VCO_2_-Total are shown in Figure 2. In males, VE/VCO_2_-slope (26.1 ± 2.0) was significantly higher than VE/VCO_2_-Nadir (24.5 ± 2.0, *p* < 0.001) and lower than VE/VCO_2_-Total (27.3 ± 2.2, *p* < 0.001). The same relationship was observed among females, where VE/VCO_2_-slope (27.7 ± 2.6) was significantly higher than VE/VCO_2_-Nadir (26.2 ± 2.4, *p* < 0.001) and lower than VE/VCO_2_-Total (28.7 ± 2.7, *p* = 0.043). Between sexes, female athletes observed higher VE/VCO_2_-slope (*p* < 0.001), VE/VCO_2_-Nadir (*p* < 0.001), and VE/VCO_2_-Total (*p* = 0.001). In univariable analysis, VE/VCO_2_ increased with age: VE/VCO_2_-slope (β = 0.093, *p* = 0.27), VE/VCO_2_-Nadir (β = 0.127, *p* = 0.14), and VE/VCO_2_-Total (β = 0.066, *p* = 0.44). 

### 3.2. Validity of VE/VCO_2_-Slope Predictions 

The accuracy of prediction equations stratified by sex is presented in Table 3. Predicted and observed values differed significantly in 9 from 12 models. Models underestimated the VE/VCO_2_-slope from −0.5 for the Loe et al. [20]. model in females up to −3.6 for the females’ formula by Ashikaga et al. [19] and the general formula by Salvioni et al. [8]. Only the model for males by Loe et al. [20] overestimated predictions, by +0.2. RMSE ranged from 2.0 to 4.5, while MAPE varied from 6.3% to 13.0%. Alignment was poor, with all ICC_3,1_ far under 0.5. Equations explained a low amount of variance, with R^2^ ranging between 0.003 and 0.031. The model’s agreement is visualized with Bland–Altman plots in Figure 3.

## 4. Discussion 

We conducted a comprehensive analysis of VE/VCO_2_ among athletes with above-average endurance levels. We examined the perspectives of estimating the important indicator of cardiorespiratory health. Our main findings are as follows: (1) prediction equations are not transferable for VE/VCO_2_-slope and do not allow for precise estimation in the athletic group, (2) predicted VE/VCO_2_-slope shows strong variability in healthy participants, and (3) several measuring methods (-slope, -Total, and -Nadir) are different both for female and male endurance athletes. 

Endurance athletes are a unique population [13]. They differ from general, nonathletic individuals in the majority of cardiorespiratory indices [34]. Physical activity provides a brilliant effect on health, but demanding endurance training could even raise the risk of CVD [5]. To mention only some differences from the untrained population, endurance athletes have higher cardiac output, their heart recovers faster, arterial hypoxemia occurs more often, and the breathing reserve is lower [34]. However, it is unknown what adaptive mechanism leads to elevation in VE/VCO_2_ among healthy athletic populations [34].

The variability of VE/VCO_2_ in endurance athletes and its indirect estimations are an understudied area. Our study filled this research gap and supplemented the knowledge of calculating bivariate cardiac indicators, especially for young adult endurance athletes. Variation associated with using prediction equations would allow a physician to compare how close the VE/VCO_2_-slope of endurance athletes is to normal reference individuals. The advantage of this study is the unique reference sample, with above-average physical levels and free of health-disturbing factors. Additionally, this is the first study that externally validated current prediction equations. 

Assuming a normal response to exercise, the VE/VCO_2_ obtained from the full CPET should be higher and nonlinear. Previous studies derived regression prediction models up to VT1 [16,35]. Our results confirmed this physiological difference in healthy endurance athletes (VE/VCO_2_-slope = 26.8 ± 2.4 vs. VE/VCO_2_-Total = 28.0 ± 2.5). VE/VCO_2_ usually grows with age. This trend is observed in untrained, healthy subjects and those with CVD [8,10]. We also noted that VE/VCO_2_ increased with age in athletic subjects independently from sex (β = 0.066–0.127). This relationship with age was the strongest for VE/VCO_2_-Nadir (β = 0.127, *p* = 0.14). However, it was not significant in either VE/VCO_2_-slope, VE/VCO_2_-Nadir, or VE/VCO_2_-Total. Some explanation is provided by the work of Salazar-Martinez et al. [23]. In their study, the VE/VCO_2_-slope was slightly higher in older endurance athletes (25.6 ± 3.7) compared to younger ones (24.3 ± 3.8). However, the authors noted that this difference was not significant between endurance athletes at the age of 16–25 years old and those at the age >45 years old (*p* = 0.146) [23]. Our study provided a wider view of other types of VE/VCO_2_, i.e., VE/VCO_2_-Nadir and VE/VCO_2_-Total. 

Derivation studies reported a low explanation of VE/VCO_2_-slope by covariates expressed as R or R^2^. Loe et al. reported R^2^ = 0.08 [20], while Salvioni et al. reported R = 0.303 for the general model, R = 0.192 for females, and R = 0.371 for males [8]. In external validation, we performed a three-step analysis. Firstly, we checked how the models for VE/VCO_2_-slope transfer above the original population. We selected a unique population—healthy, free of disturbing comorbidities, and with high endurance capacity. Based on previous studies on athletic cohorts for the remaining CPET variables, we stipulated that there would be an underestimation [13,14,21]. Reduced calculations are more often noticed among physically active subjects than in general groups [14,21]. Overall, equations provided values from −3.6 for Ashikaga et al. (in females) [19] and Salvioni et al. (general model) [8] to +0.2 for Loe et al. (in males) [20]. Differences were large enough that they reached significance in 9 of 12 models (75%). A transfer error of 6.3–13.0% in our athletic cohort is lower than that reported by Salvioni et al. (approximately 24.0%) when transferring from the general to clinical sample [8]. Some clarification was reported by Petek et al., who noted that endurance athletes could continue the physical effort longer and with higher intensity. Thus, their VE/VCO_2_-slope increases significantly [3]. Even up to 33% of them could exceed the recommended normal cutoff of 30, while up to 13% could fall beyond 34 [3]. However, our subjects have only slightly elevated VE/VCO_2_: the majority of them fell within the recommended ranges, and only four (6.3%) females exceeded the cutoff of 34 for VE/VCO_2_-Total. Even though some of our athletes fall above the recommended cutoff, the average VE/VCO_2_-slope was comparable to those observed by Brown et al. in juvenile cyclists (26.8 ± 2.4 vs. 28.14 ± 3.89) [22].

Secondly, we aimed to evaluate how the predicted slopes explained variance in the observed measures. To verify deeper relationships in the VE/VCO_2_ course, we enriched our analysis by regressing predictions against directly measured slopes. R^2^ ranged between 0.003 and 0.031 (0.03–3.1%). We noted that the trend of predictions is generally consistent; however, the covered amount of variance was still poor. 

Finally, by low ICC, we noted that VE/VCO_2_ variability remains large, and agreement was low between predicted and observed measures (all ICC < 0.5) [32,33]. The least accurate was a general model by Salvioni et al. [8]. Interestingly, the general model by Sun et al. [10] showed the highest accuracy. A plausible explanation could be that it was because that model was derived from a numerous, reliable population of 474 individuals with varied fitness levels. Moreover, an additional advantage of the Sun et al. [10] model is the inclusion of body height as a covariate, which could add to the prediction value. For precise descriptions of all selected models, see Table 2. To summarize, the VE/VCO_2_-slope may have good visit-to-visit repeatability in a single athlete [22]. However, it is poorly transferable between populations. This observed variability in VE/VCO_2_-slope predictions could emerge from improved breathing control with regular physical training [24]. 

### 4.1. Practical and Clinical Implications

Knowledge of the patient’s endurance capacity remains important in the athletic assessment [6]. Our findings enable the evaluation of VE/VCO_2_ as a factor related to cardiovascular functions and facilitate clinical decision making. Results can be a valuable supplement to the CPET assessment. Additionally, we confirmed that slightly elevated VE/VCO_2_ can also be observed among younger individuals during strenuous physical effort, especially females. It is not unusual for VE/VCO_2_ values to be increased in endurance athletes during efforts above VT1, especially at the maximal levels. Our findings are in line with the report by Petek et al. [3] and correspond with the results of Brown et al. [22]. Increased VE/VCO_2_ should be an indicator for further, more precise medical evaluation to ensure safe physical activity.

Another practical implication of this study is the assessment of how the well-trained participant fits into the reference values [5]. Further determination of the degree of cardiorespiratory weakness or abnormalities is facilitated. What is more, we see that prediction equations could be used only to supplement the comprehensive diagnostic process, but not to make a definitive diagnosis. Comparing all three measurements provides a more comprehensive picture of cardiorespiratory health [16]. If one type of VE/VCO_2_ is elevated, the physician could recalculate the remaining two to avoid over-awareness and misdiagnosis [16].

The present research confirms that reference values have limited repeatability between populations. The universal use of prediction equations for VE/VCO_2_-slope between athletic and untrained populations is not recommended. In general, the prediction equations for VE/VCO_2_-slope explained only minimal individual variance. The underlying reason should be confirmed by future studies. However, we stipulate that training specificity or additional somatic indices could individually contribute to the variability in predictions. All of the variance cannot be explained by available simple models, at least in endurance athletes. 

Moreover, we underline that the most reliable and repeatable VE/VCO_2_ index is the Nadir. It is calculated from a precisely defined time interval, so it avoids the subjective susceptibility of the VT1 assessment or the athlete’s motivation to push to maximum effort [36]. Nevertheless, our results facilitate the selection of the most accurate equation for indirect prediction when full CPET is unavailable. 

### 4.2. Limitations and Interpretation

The investigated population in the majority consisted of younger individuals. However, our age distribution fitted in the ranges of the original studies. Our cohort meets the required size to ensure reliable results, but previous models were developed based on broader samples. This study also guides the thoughts to reconsider the VE/VCO_2_ in other populations, e.g., master athletes. Our testing center does not provide lactate measurements. Thus, all the thresholds were assessed based on the gas exchange responses. It should be acknowledged that ethnicity could influence endurance capacity because all of our athletes were Caucasians [37]. We did not consider the training period of our participants. However, assuming their high fitness level, the performance of enrolled athletes should not vary significantly over the season [38].

Our group consisted of healthy individuals with a high fitness level. Above-average endurance of our athletes should be considered when interpreting the results of the present study. According to our knowledge, it is the first and largest validation study to date comparing predicted and observed VE/VCO_2_-slopes in the athletic population. Independent replication on other populations, including clinical samples, would be intriguing.

### 4.3. Further Research Directions

Original derivation studies merge different testing protocols and modalities. The type of CPET could significantly contribute to the final measurements [16,39]. Novel, more advanced models should be created under unified CPET conditions and report testing modality. Additional confirmation for groups with more varied age distribution, including pediatric and master athletes, is always welcome. It is worth underlining that such variability between the observed and predicted VE/VCO_2_-slope occurs in a healthy population. We stipulate that differences could become even more significant among clinical conditions when other comorbidities emerge [40]. Perhaps, supplementing future models with other covariates would enrich their accuracy. Moreover, clinical guidelines should consider methods of calculation and distinguish between athletic and nonathletic individuals. Future research on other sports disciplines should be also conducted. 

## 5. Conclusions

VE/VCO_2_-slope, VE/VCO_2_-Nadir, and VE/VCO_2_-Total significantly differed in young endurance athletes. The differences were stronger in male than female endurance athletes. Female endurance athletes observed higher VE/VCO_2_ than male endurance athletes in all measuring methods. VE/VCO_2_-slopes were significantly downgraded by 11 from 12 predicted equations. Prediction inaccuracies were higher in males than in females. Indirect calculations are not transferable for VE/VCO_2_-slope between trained and untrained individuals. Estimated VE/VCO_2_-slopes should be carefully used for clinical practice and sports diagnostics. Physicians should take care to properly assess cardiorespiratory responses to exercises in specific populations.

## Figures and Tables

**Figure 1 jcm-13-00490-f001:**
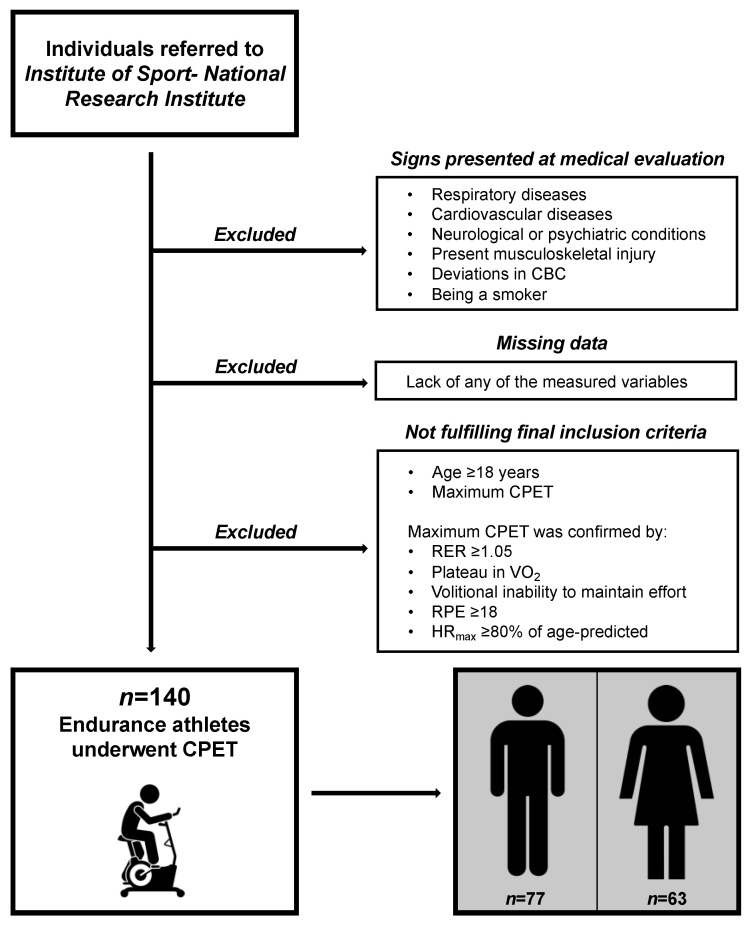
Visual presentation of participant recruitment procedures. CBC, complete blood count; CPET, cardiopulmonary exercise test; RER, respiratory exchange ratio; VO_2_, oxygen uptake; RPE, rating of perceived exertion; HR_max_, maximal heart rate.

**Figure 2 jcm-13-00490-f002:**
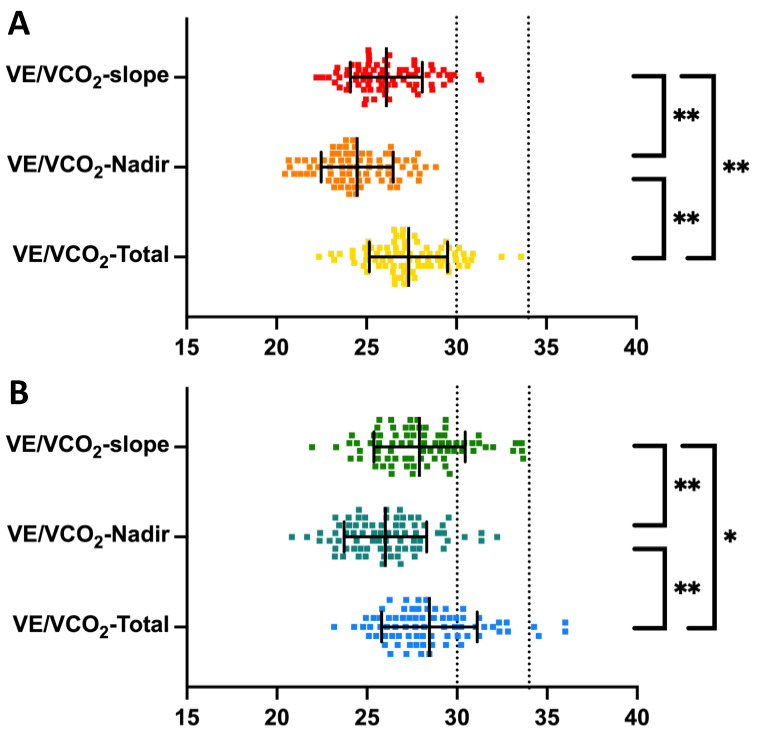
Interdependency in methods measuring ventilatory efficiency among endurance athletes. VE/VCO_2_-slope, ventilatory efficiency from start to the first ventilatory threshold; VE/VCO_2_-Nadir, the lowest 30 s continuous average for ventilatory efficiency; VE/VCO_2_-Total, ventilatory efficiency from start to peak effort. Panel (**A**) illustrates a comparison for males, and panel (**B**) illustrates a comparison for females. Error bars present mean with standard deviation. Black dotted lines represent the recommended cutoffs of 30 and 34. Significant differences with *p* < 0.001 are marked with (**) and significant differences with *p* < 0.05 are marked with (*).

**Figure 3 jcm-13-00490-f003:**
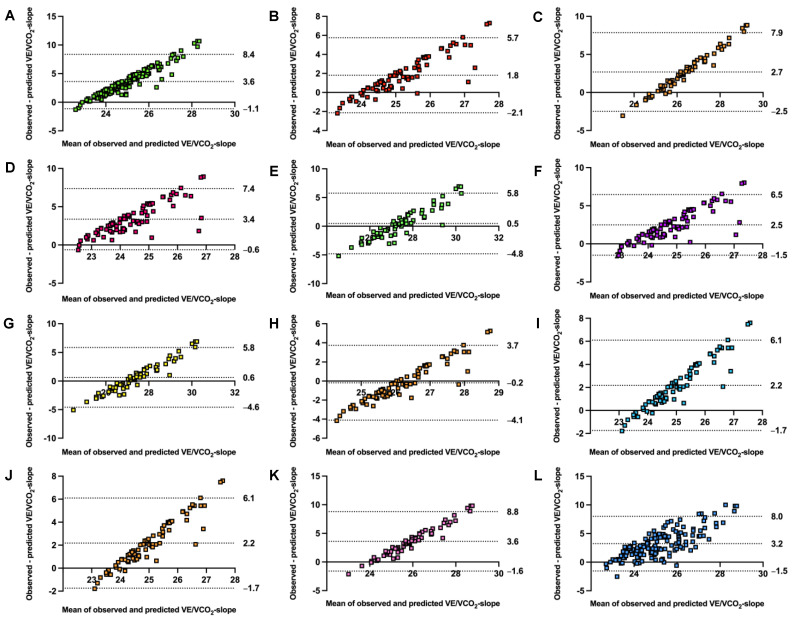
Bland–Altman plots comparing observed with estimated ventilatory efficiency slope by validated models. VE/VCO_2_-slope, ventilatory efficiency slope from start to first ventilatory threshold. Panel (**A**) general model by Salvioni et al. [8], panel (**B**) male model by Salvioni et al. [8], panel (**C**) female model by Salvioni et al. [8], panel (**D**) male model by Kleber et al. [17], panel (**E**) female model by Kleber et al. [17], panel (**F**) male model by Neder et al. [18], panel (**G**) female model by Neder et al. [18], panel (**H**) male by Loe et al. [20], panel (**I**) female by Loe et al. [20], panel (**J**) male by Ashikaga et al. [19], panel (**K**) female model by Ashikaga et al. [19], and panel (**L**) general model by Sun et al. [10]. All models were validated for the VE/VCO_2_-slope from the start to the first ventilatory threshold. The central dotted line represents mean bias. The upper and lower dotted lines represent 95% confidence intervals.

**Table 1 jcm-13-00490-t001:** Participant characteristics.

Variable		Total (*n* = 140)	Sex	
			Females (*n* = 63)	Males (*n* = 77)
**Age (years)**		22.7 ± 4.6	23.8 ± 4.2	21.8 ± 4.8
**Height (cm)**		174.8 ± 9.9	166.3 ± 6.2	181.6 ± 6.3
**Weight (kg)**		69.3 ± 10.1	61.0 ± 5.5	76.1 ± 7.6
**BMI (kg·m^−2^)**		22.6 ± 1.7	22.1 ± 1.6	23.1 ± 1.7
**Primary sport**	**Speedskating**	59 (42.1)	26 (41.3)	33 (42.9)
	**Triathlon or cycling**	56 (40.0)	30 (47.6)	26 (33.8)
	**Other**	25 (17.9)	7 (11.1)	18 (23.3)
**HR (beats·min^−1^) **		190.9 ± 8.9	191.0 ± 9.1	190.8 ± 8.7
**VE (L·min^−1^)**		154.5 ± 34.1	127.8 ± 21.1	176.3 ± 26.3
**VO_2_peak (L·min^−1^)**		3.86 ± 0.82	3.21 ± 0.48	4.40 ± 0.64
**VCO_2_ (L·min^−1^)**		4.36 ± 0.96	3.57 ± 0.52	5.00 ± 0.73
**VO_2_peak (mL·kg^−1^·min^−1^)**		55.2 ± 8.6	52.1 ± 7.0	57.8 ± 9.0
**RR (breaths·min^−1^)**		60.0 ± 7.6	60.2 ± 6.7	59.9 ± 8.3
**VT (L)**		2.81 ± 0.64	2.30 ± 0.32	3.22 ± 0.53
**RER (VO_2_/VCO_2_)**		1.14 ± 0.05	1.13 ± 0.05	1.15 ± 0.05
**VE/VCO_2_-slope**		26.8 ± 2.4	27.7 ± 2.6	26.1 ± 2.0
**VE/VCO_2_-Nadir**		25.2 ± 2.3	26.2 ± 2.4	24.5 ± 2.0
**VE/VCO_2_-Total**		28.0 ± 2.5	28.7 ± 2.7	27.3 ± 2.2
**O_2_P (VO_2_/HR)**		20.7 ± 4.4	17.3 ± 3.0	23.5 ± 3.3
**Testing duration (minutes)**		21.3 ± 2.6	21.1 ± 2.7	21.4 ± 2.6
**Workload (watts)**		320.4 ± 76.2	266.7 ± 40.8	364.4 ± 70.0

BMI, body mass index; HR, peak heart rate; VE, peak minute ventilation; VO_2_peak, peak oxygen uptake; VCO_2_, peak carbon dioxide output; RR, peak respiratory rate; VT, tidal volume; RER, peak respiratory exchange ratio; VE/VCO_2_-slope, ventilatory efficiency from start to the first ventilatory threshold; VE/VCO_2_-Nadir, the lowest 30-s continuous average for ventilatory efficiency; VE/VCO_2_-Total, ventilatory efficiency from start to peak effort; O_2_P, peak oxygen pulse. Data are presented as mean ± standard deviation for continuous variables or number (percentage) for categorical variables.

**Table 2 jcm-13-00490-t002:** Prediction equations selected for validation.

Reference	Model		Testing Protocol	Sample Size (Total/Males/Females)	Age (Years)
	Males	Females			
**Salvioni et al. [8]**	20.227 + 0.095 · age	23.808 + 0.052 · age	Cycling CPET; ramp protocol. Running CPET; Bruce protocol.	1136/773/363	13–83
	21.413 + 0.08 · age				
**Kleber et al. [17]**	19.9 + 0.13 · age	24.4 + 0.12 · age	Running CPET; modified Naughton protocol with increases in gradient and speed of 1 MET every 2 min.	101/45/56	16–75
**Neder et al. [18]**	21 + 0.12 · age	25.2 + 0.08 · age	Cycling CPET; ramp protocol with increases in power of 10–25 W·min^−1^ in females and 15–30 W·min^−1^ in males.	120/60/60	20–80
**Loe et al. [20]**	23.897 + 0.072 · age + 0.826	25.549 + 0.072 · age	Running CPET; ramp protocol with increase in speed of 1 km·h^−1^ or gradient of 2% every 2–3 min.	4631/2261/2370	20–90
**Ashikaga et al. [19]**	22.4 + 0.07 · age	22.467 + 0.07 · age	Cycling CPET; ramp protocol with increases in power of 10 W·min^−1^ or 20 W·min^−1^.	529/274/255	20–78
**Sun et al. [10]**	34.38 + 0.082 · age − 0.0723 · height		Running or cycling CPET; incremental maximal protocols with varying duration.	474/310/164	37–74

CPET, cardiopulmonary exercise test; MET, metabolic equivalent. For all models, age is expressed in years. All models apply to the ventilatory efficiency slope from the start to the first ventilatory threshold.

**Table 3 jcm-13-00490-t003:** Validity of ventilatory efficiency slope predictions.

Prediction Equation	Predicted VE/VCO_2_-Slope	Difference	RMSE (%RMSE)	MAPE	*p*-Value	ICC_3,1_ (95% CI)	R^2^
Salvioni et al. (general model) [8]	23.2 ± 0.4	−3.6 ± 2.4	4.3 (16.2)	13.0	**<0.001**	0.028 (0.002, 0.053)	0.009
Salvioni et al. (males) [8]	24.3 ± 0.4	−1.8 ± 2.0	2.7 (10.3)	7.5	**<0.001**	0.043 (0.017, 0.069)	0.010
Salvioni et al. (females) [8]	25.1 ± 0.2	−2.7 ± 2.6	3.8 (13.5)	10.1	**<0.001**	<0.001 (−0.026, 0.026)	0.004
Kleber et al. (males) [17]	22.7 ± 0.6	−3.4 ± 2.0	3.9 (15.1)	12.5	**<0.001**	0.056 (0.031, 0.082)	0.010
Kleber et al. (females) [17]	27.3 ± 0.5	−0.5 ± 2.7	2.7 (9.8)	7.5	0.44	<0.001 (−0.026, 0.026)	0.004
Neder et al. (males) [18]	23.6 ± 0.6	−2.5 ± 2.0	3.2 (12.3)	9.4	**<0.001**	0.052 (0.027, 0.078)	0.010
Neder et al. (females) [18]	27.1 ± 0.3	−0.6 ± 2.6	2.7 (9.8)	7.5	0.20	<0.001 (−0.026, 0.026)	0.004
Loe et al. (males) [20]	26.3 ± 0.3	+0.2 ± 2.0	2.0 (7.6)	6.3	**0.03**	0.036 (0.010, 0.062)	0.010
Loe et al. (females) [20]	27.3 ± 0.3	−0.5 ± 2.6	2.7 (9.6)	7.4	0.33	<0.001 (−0.026, 0.026)	0.004
Ashikaga et al. (males) [19]	23.9 ± 0.3	−2.2 ± 2.0	2.9 (11.3)	8.5	**<0.001**	0.033 (0.007, 0.059)	0.010
Ashikaga et al. (females) [19]	24.1 ± 0.3	−3.6 ± 2.6	4.5 (16.1)	12.6	**<0.001**	<0.001 (−0.026, 0.026)	0.003
Sun et al. (general model) [10]	23.6 ± 0.9	−3.2 ± 2.4	4.0 (15.1)	11.7	**<0.001**	0.112 (0.087, 0.138)	0.031

VE/VCO_2_-slope, ventilatory efficiency slope from start to the first ventilatory threshold; RMSE, root mean square error; %RMSE, percentage root mean square error; MAE, mean absolute error; MAPE, mean absolute percentage error; ICC_3,1_, two-way mixed effect interclass correlation coefficient; CI, 95% confidence interval; R^2^, adjusted coefficient of determination. The observed VE/VCO_2_-slope was 27.7 ± 2.6 for females, 26.1 ± 2.0 for males, and 26.8 ± 2.4 for the total population. %RMSE was calculated by dividing RMSE by the mean of the observed VE/VCO_2_-slope. Significant *p*-values (<0.05) are bolded. Negative values (−) of difference mean underestimation, i.e., predicted values were lower than observed values. Positive values (+) mean overestimation, i.e., predicted values were higher than observed values.

## Data Availability

The raw data supporting the conclusions will be made available on a reasonable request to the corresponding author.

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
