# Peer review of "Is the Ventilatory Efficiency in Endurance Athletes Different?—Findings from the NOODLE Study"

_jcm, 2024, doi:10.3390/jcm13020490_

Round 1

Reviewer 1 Report

Comments and Suggestions for Authors

In general, the study has been carefully prepared and I think it will contribute to the literature. The visual is used in its place. The statistical analyses of the study were done in a remarkable way. Materials-Methods and Findings have been prepared meticulously. There is no deficiency in many aspects of the study. However, English grammar support can be obtained in some sections. I have indicated below the places I want to be explained. 

Title: Could the title of the study be more striking? This suggestion is optional. 

42-55. The English language should be revised in this section. 
124. Which tool did you use in the measurements? Is it possible to specify?

140. From which branch these athletes were selected, I think it should be briefly explained.

Table 1. If these are standard deviations, I think it would be better to put the icon (±) in front of it.

369-372. (The English language should be revised in this section.)

5. Conclusion (In this section English grammar should be reorganised.)

Comments on the Quality of English Language

English grammar support can be obtained in some sections.

Author Response

The authors would like to thank for devoting time to analyze our manuscript and for all the suggestions, which contributed to making it more valuable. Your feedback and precise review have enhanced the quality of the manuscript. Moreover, the Authors are grateful for seeing the value of our manuscript.

Revisions were marked with the tracking-changes mode. Moreover, we applied English proofreading according to your suggestion. We did our best to revise our manuscript according to your comments.

We present our reply point-by-point below:

•Title: Could the title of the study be more striking? This suggestion is optional. 

Authors’ reply: We revised our title to present it in a more interesting form.
•42-55. The English language should be revised in this section. 

Authors’ reply: We revised our English with the help of a Native Speaker as you recommended. We have rewritten several sentences in lines 42-55 to ensure that the text is clear and concise.

•124. Which tool did you use in the measurements? Is it possible to specify?

Authors’ reply: Thank you for your comment. We have described in more detail used equipment.
•140. From which branch these athletes were selected, I think it should be briefly explained.

Authors’ reply: Thank you very much for this valuable comment. We agree that this issue requires additional information. We enriched our text with participants’ primary disciplines and their sports characteristics. The information was included in section 2.1 Study design and 2.4 Sample characteristics. Moreover, we add this issue in the Table 1.
•Table 1. If these are standard deviations, I think it would be better to put the icon (±) in front of it.

Authors’ reply: Of course, the use of brackets “()” could be confusing. We switched to the “±” format in the Table 1. We also revised the “±” in other tables and in the text.
•369-372. (The English language should be revised in this section.)

Authors’ reply: Yes, we agree with you. We revised our language in this paragraph.
•5. Conclusion (In this section English grammar should be reorganised.)

Authors’ reply: Thank you for a valuable suggestion. We applied deep English proofreading to ensure maximal clarity of the text. We also enriched this section with additional information.

Once again, the Authors would like to thank for your crucial contribution to the article and for seeing the value of our work. We are sorry for any inconveniences you could have experienced which resulted from our previous inaccuracies. We would be grateful if our manuscript in its current form would fulfill the requirements of the Journal of Clinical Medicine. We did our best to revise our manuscript according to your comments.

Reviewer 2 Report

Comments and Suggestions for Authors

This paper is a well-performed study within the journal's scope and relevant to medical professionals and the use of testing to evaluate cardiorespiratory health. The manuscript and this method study should be particularly interesting to readers working in this specific domain and researchers. The manuscript is well-written, and the authors' arguments and constructs are possible follow, but a bit demanding if you are not a specialist in the topic. This study was aimed to externally validate prediction equations for the VE/VCO2-slope in the athletic population and explain the relationship between the VE/VCO2-slope, VE/VCO2-Nadir, and VE/VCO2-Total. The author presents a novel data set on Endurance trained athletes on the variability of the VE/VCO2 in endurance athletes and its indirect estimations compared to the existing research on the topic.

However, prior to publication, I have some suggestions that the authors should consider (see attached file).

Author Response

The Authors would like to thank for devoting time to analyze our manuscript and for all the suggestions, which contributed to making it more valuable. Your feedback and precise review have enhanced the quality of the manuscript. Moreover, the Authors are grateful for seeing the value of our manuscript. We agree with all your suggestions. Revisions were marked with the tracking-changes mode. We did our best to revise our manuscript according to your suggestions.

We present our reply point-by-point below:

  • Line 26-29 I don’t see how you can or should conclude like this? If we first need to test fitness level (consider as you put it) in EAs before testing VE/VCO2 to evaluate the risk of HF, I think we already have taken the risk? Correct me if I have misunderstood, isn’t the idea that we could use the VE/VCO2 testing at submaximal loads to predict cardiovascular diseases (and exercise response)? I think that prediction equations for VE/VCO2-slope performed accurately in EA, but it is different from ordinary subjects shown by the literature? My suggestion is that you focus your conclusions more on answering your two aims and be broader about how practitioners should use fitness level and VE/VCO2 under the section on practical implications (because this doesn’t become clear to me).

Authors’ reply: Thank you for this comment. It could be quite confusing. We clarified all raised areas and we have rewritten our conclusions both in the abstract and in the manuscript to properly answer the study aims. Moreover, we expanded and clarified section 4.1 Practical and clinical implications.

  • Line 30-31. Don’t repeat the words from the heading of the paper (Ventilator efficiency, Endurance athletes); this does not increase the number of readers that search and find your article in the databases.

Authors’ reply: We revised our keywords as you recommended. We switched to other sets of keywords to avoid repeating the title.

Introduction

  • Line 51. HF is an abbreviation that is not explained before.

Authors’ reply: We explained the abbreviation “HF” as you suggested.

  • Line 51-55. From this perspective, and that also EA (unexpected) gets cardiorespiratory diseases or HF, isn’t it convenient to enlighten the lack of research on this topic here (the findings in your data are on EA and question the cut of values for EA.)? I see it is presented in the sections below but consider a paragraph about it after line 55 and remove it from the section below.

Authors’ reply: Thank you for this valuable comment. We enlighten the lack of research on the athletic population in this area as you recommend. Moreover, we transfer the parts of the text included below after line 55.

  • Line 54-55. high risk of?

Authors’ reply: We meant the “high risk of mortality” as presented in the cited references. We add this term to the text.

  • General the introduction seams fine, however there are some information in the beginning of the discussion who can be included in the introduction and deleted it there (in the section Line 283-191). For me personally as a reader without medicine education the concepts/problems around prediction, underestimate and overestimate might have been explained better.

Authors’ reply: We transferred the text from lines 283-291 into the Introduction. We agree with your comment. Moreover, we briefly explained the concept around prediction, under- and over- estimation in lines 75-87.

Method

  • Line 85. Delete “Being”

Authors’ reply: We improved as you recommended.

  • Line 98-100 Punctuation: For the visual presentation of the recruitment procedures, see Figure 1, and for exact definitions of exclusion criteria, see Supplementary Material (Table S2).

Authors’ reply: We also improved as you recommended.

  • Line 104. No need for writing Abbreviation: Revise under all fig and tab.

Authors’ reply: We revised the term “Abbreviations” under all figures and tables.

  • Line 110-113. I am used to stronger criteria for reaching VO2max (RER ≥1,08, Borg≥19, and a HR≥95%) so you should use VO2peak or give a valid reference to the values you have used.

Authors’ reply: Thank you for such a valuable comment. According to your recommendation, in all places of the manuscript we use the term “VO2peak”. Moreover, we cite the references where those values were used.

  • Line 125. What kind of HR monitor was used, was it connected to the Metamax or?

Authors’ reply: We used the Polar H10 chest strap (Polar Electro Oy, Kempele, Finland) to monitor heart rate. It was continuously synchronized with the Cortex B3 Metamax. We add these issues in the text.

  • Line 130. Is there some validation study on this Cortex B3 Metamax?

Authors’ reply: There are validation studies on Cortex B3 Metamax by Vogler et al. (DOI: 10.1080/02640410903582776).

  • 2.4 sample characteristics: Congrats, One hundred forty athletes is quite an impressive data

collection and a lot of work.

Authors’ reply: The Authors are grateful for seeing the value of our work!

  • Line 143-148. I don’t know, and it may be out of the scope of the article but those who have been identified as over the cut-off values, did you offer them some following up by professionals? If so, it could be mentioned here.

Authors’ reply: We did not followed-up those participants after the study. However, to ensure their safety, we provided a health assessment by the Medical Doctor before the exercise tests. We will consider follow-up of these endurance athletes in the future as a topic for the next studies. 

  • Table 1.

Replace “VO2” with VO2max or VO2peak

Authors’ reply: We improved as you recommended.

  • Line 160. Replace “Takkken” with Takken

Authors’ reply: We improved as you recommended.

  • Table 2. I don’t know if this table needs to be in a format that fits the boundary the rest of the text? Check the journals guidelines.
  • Figure 2. format that fits the boundary?
  • Table 3. format that fits the boundary?

Authors’ reply: We check the guidelines of the Journal of Clinical Medicine. It is not necessary to provide tables in a format that fits the boundary of the text. Before final publication, the manuscript underwent Author’s Proofreading when the Layout Team revise all the tables according to journal standards.

  • Line 175- and how was the distribution?

Authors’ reply: Thank you for this suggestion. It was a parametric (normal) distribution. We add this information to the text.

Results

  • Line 254. It is a bit difficult for me to understand that this is an underestimate or an overestimate? Correct me if I am wrong but to this variation (lack of precision) associated from using reference values/tables, the reference tables would only allow a CPX administrator to say how close to “normal” the EA/patient’s VʹE/VʹCO2 slope value is?

Authors’ reply: We agree with your comment. The existing footnote under the Table 3 could be confusing. We clarified what underestimation and overestimation mean to ensure a unified data presentation. Moreover, as we agree with your comment, we add this statement in lines 344-346.

Discussion

  • Line 277. Yes, but only for EA young adults (22.7 years ±4.6)

Authors’ reply: Yes, of course. We add this to the text to ensure proper interpretation.

  • Line 283-288. This information should be presented in the introduction, and it would be easier for the uninitiated in the topic to read.

Authors’ reply: We agree with your comment. We transferred this part into the Introduction.

  • Line 293. I cannot see that your data is supporting this, or do you claim that others have shown that VE/VCO2 is age related? Your athletes were 22.7 years old ±4.6.

Authors’ reply: Of course, it could be quite confusing in the previous form. Other studies showed that ventilatory efficiency is age-related. Despite, the average age of the study population is 22.7 years old, we noticed a similar trend. We have rewritten this part and added the necessary information from other studies to ensure a proper understanding of the text. 

  • Line 294. What underlying relationship are you referring to? Does this indicate that VE/VCO2-nadir is less sensitive or the others more sensitive???

Authors’ reply: We thought about the relationship presented in other studies on the untrained, general population. Previous studies found that VE/VCO2-slope increases with age. Our study adds that this is consistent for all the measuring types (also VE/VCO2-Nadir and VE/VCO2-Total). We add a precise description in the text to avoid any misunderstanding. 

  • Line 296. Some explanations is provided….which ones?

Authors’ reply: We also add the precise description of the explanation provided by Salazar Martinez et al. in their study. Moreover, we emphasized how their clarification relates to our findings.

  • Line 332. Replace “Perhaps” with A plausible explanation could be….

Authors’ reply: We improved as you recommended.

  • Line 337. Replace "Perhaps" with This observed ….

Authors’ reply: We improved as you recommended.

General to the discussion

  • The whole discussion brings up a lot of “whys that” when I read it. You have investigated EA, and I am missing some explanations of why they differ from the rest of the population; why is there a difference between sexes (is it only because they are at a different fitness level, or is it sex-related)? Is there a different cut of fore EAs?

Further, when I quickly examine some of the published literature on the VE/VCO2 slope, it seems to me that the result differs from paper to paper due to age, sex, BMI, methods/protocols, treadmill running, bicycle ergo, and different lifestyle issues. Considering all the data presented in your paper and in others' work, only some actually measure the VE/VCO2 slope above the cut-off value (your data and data in Fig 3, and others), except for investigations on older subjects. Is there something special about the population participating in this kind of experiment, or isn't this measure valid enough?

Or is it me who doesn't understand the concept ?

However, I suggest you add more discussion to AE, why they differ, and how these methods can best be used on EAs. You have reported sex differences; why is there a difference? You have found significant differences between VE/VCO2 slope-nadir and total, why?, what is the practical significance? And what do you recommend should be used?

Authors’ reply: Thank you for this comment. We add a precise description of why endurance athletes are unique and different. We also clarified what is special about the population participating in our study. All the details were provided in the new paragraph in lines 333-339. Moreover, we support the text with additional references.

The difference between males and females is sex-related. Currently, there are no different cut-off values for endurance athletes. However, we underlined the need to consider whether the participant is an endurance athlete or not in the clinical guidelines (see lines 476-477).

We also enriched section 4.1 Practical and clinical implications. We provided the information with practical significance and recommendations as you suggested.

  • Line 364. Replace “it” with It.

Authors’ reply: We improved as you recommended.

  • Line 346. Strenuous physical activity, > VT1? Is it Strenuous?

Authors’ reply: Thank you for this comment. We described more precisely what we considered as strenuous physical activity.

  • Line 384-386. Can this be a part of the conclusions, it does not fit under the heading of the

section?

Authors’ reply: You are right. We transferred this part to the Conclusions.

Conclusions

  • Do you answer your aims?

Line 70-73. This study aimed to (1) externally validate prediction equations for the VE/VCO2-slope in the athletic population and (2) explain the relationship between the VE/VCO2-slope, VE/VCO2-Nadir, and VE/VCO2-Total.

Line15-16 . in the abstract: (1) investigate the relationship between different methods of

calculation of VE/VCO2 and (2) externally validate prediction equations for VE/VCO2.

Here is your conclusion from line 390-395

VE/VCO2-slopes were significantly downgraded in endurance athletes by 11 from 12 predicted equations. Predictions are not transferable for the VE/VCO2-slope between trained and untrained populations. Indirect estimations should be carefully used for clinical practice and sports diagnostics. VE/VCO2-slope, -Nadir, and -Total were significantly different in the athletic population. Physicians should be acknowledged to properly assess cardiorespiratory responses to exercises.

Provide a broader explanation for aim 2!

Should the sex difference should be mentioned?

The dissonance between the different sections makes the last part of the paper a bit challenging to follow. Ensure that the aims and conclusions in the abstract, introduction, and conclusion are better related.

Good look with your further work!

Authors’ reply: Thank you for paying attention to details. The previous form of aims and conclusions in the abstract and the text could be ambiguous. We applied deep revision for both. The goals of this study and conclusions are clearer and more concise in their current form. We provided a broader explanation for aim 2 and mentioned the sex differences.

Thank you for noticing the value of our work and all the time devoted to improve our manuscript.

Once again, the Authors would like to thank for your crucial contribution to the article and seeing value of our work. We are sorry for any inconveniences you could have experienced which resulted from previous inaccuracies. We would be grateful if our manuscript in its current form would fulfill the requirements of the Journal of Clinical Medicine. We did our best to revise our manuscript according to your comments.